# OpenReview forum: "RFG: Test-Time Scaling for Diffusion Large Language Model Reasoning with Reward-Free Guidance"
_ICLR.cc/2026/Conference — Submitted to ICLR 2026_

### Official Review · Reviewer_JGzp · 2025-10-28

**Soundness:** 2
**Presentation:** 3
**Contribution:** 2
**Rating:** 4
**Confidence:** 4

**Summary:**

This paper studies the techniques of test time scaling for diffusion language models under the guidance of process rewards. To parameterize process rewards, this paper proposes using log-likelihood ratio between enhanced and referenced diffusion language model, which results in a form aligns with classifier-free guidance. The effect is validated on several representative benchmarks on math reasoning and coding.

**Strengths:**

1. This paper presents a simple method to enhance sampling results of diffusion language models, which is theoretically identical to classifier-free guidance.
2. The method shows improvement on some representative benchmarks of math and coding and different models. This confirms the generality and effectiveness of the proposed method.
3. It is interesting to see the close connection between implicit reward modeling through likelihood ratio and classifier-free guidance. This contribution may open possibility for future research on the alignment of diffusion models.

**Weaknesses:**

1. The method requires to maintain two diffusion language models and relies on tuning a hyperparameter, which is not very convenient.
2. The paper is not self-contained enough in using likelihood ratio as the rewards. It is confused for readers less familiar about process rewards.
3. The performance gains is relatively minor and is only compared to a weak ensemble baseline.

**Questions:**

1. Given the alignment of the introduced method and classifier-free guidance, is it possible to achieve similar effect as the paper show using the log-likelihood ratio from a single diffusion lm w and w/o prompt?
2. With the implicit reward modeling shown, is is possible to provide process reward to help reinforcement learning of diffusion models?
3. What is the best of 2 in your main resutls (Tab. 1 and Tab. 2), either selecting with oracle or likelihood/confidence-based scoring. I am curious about the gap between Bo2 and RFG.

---

> ### Author Response · Authors · 2025-11-26
>
> We thank the reviewer for their review and for highlighting the "generality and effectiveness" of our method and recognizing the "interesting connection between implicit reward modeling and CFG". We also appreciate their feedback and address each concern below.
>
> ---
>
> ## **[W1] Convinience**
>
> The reviewer is concened about the convinience of RFG, saying that it "requires to maintain two diffusion language models and relies on tuning a hyperparameter."
>
> We'd like to clarify these two practical points:
>
> 1. **No Need to Train Two Models**
>
>    RFG **does not require training or maintaining any new models**.
>
>    It simply reuses existing **public checkpoints** of a post-trained model and its corresponding base model.
>
>    Since these checkpoints are already jointly released publicly, RFG can be applied **entirely off-the-shelf**, without additional data, compute, or training.
>
>    In this sense, RFG is highly practical and convinient.
>
> 2. **Performance is Robust against Hyperparameter $w$:**
>    As shown in **Fig. 3** (Fig. 2 in updated version) in Sec. 4.3, RFG exhibits a **broad, stable plateau** across a wide range of $w$ values. This indicates **very low sensitivity**, and in practice one typically tries fewer than 3 values to obtain improvements.
>
>    Thus, RFG requires *minimal tuning*.
>
> ---
>
> ## **[W2] Clarification on Likelihood Ratio as Rewards**
>
> The reviewer suggests the reward formulation may confuse readers less familiar with process rewards.
>
> We respectfully clarify that **our formulation and deduction is rigorous and self-contained**. Prop. 3.1 provides a complete derivation showing that the trajectory-level log-likelihood ratio decomposes exactly into step-wise guidance terms under the Markov structure of discrete diffusion. This result follows directly from standard probability identities and does not assume any prior knowledge of PRMs.
>
> We appreciate **the reviewer's underlying point about accessibility** and have **added a paragraph** that **provides high-level intuition and background** explaining why the log-likelihood ratio is a natural and principled choice to support readers who are less familiar with reward modeling, as the reviewer helpfully suggested.
>
> ---
>
> ## **[W3 & Q3] Performance Gains and Baselines**
>
> The reviewer states that the performance gains appear "relatively minor", mentions that RFG is only compared to an ensemble baseline and requests Best-of-2 (Bo2) comparisons.
>
> We clarify each concern as follows.
>
> ### **Significance of Gains**
>
> RFG is a **training-free** method operating solely at test time, yet it can achieve improvements that are **comparable to or exceed** those obtained from expensive post-training (e.g., GRPO). For example:
>
> - d1-LLaDA (RL-trained): +2.7% (GSM8K), +0.4% (MATH-500). **RFG**: +2.2% (GSM8K), +2.8% (MATH-500).
> - diffuCoder (RL-trained): +7.3% (HumanEval). **RFG**: +9.2% (HumanEval).
>
> For a training-free method applied on top of strong, optimized models, these improvements are substantial.
>
> ### **Rationale for Ensemble Baseline Chocie**
>
> The ensemble baseline is a **compute-matched control**, using *exactly the same two models and the same number of forward passes* as RFG at each denoising step.
>
> This makes it the most direct and fair ablation to isolate the contribution of RFG’s guidance.
>
>  RFG’s consistent and meaningful gains over this cost-matched baseline demonstrate the effectiveness of our method.
>
> ### **Bo2 Results**
>
> We thank the reviewer for the suggestion. Our Bo2 result show that
>
> - Bo2 is effectively identical to the original model’s performance.
> - Bo2 **underperforms RFG** across all evaluated models and benchmarks.
>
> This is because dLLMs use **low sampling temperature** to achive better performance (as recommended in original paper of the models), which reduces output diversity. We noticed that raising temperature increases diversity but causes a large performance drop, making Bo2 performance even worse. This means RFG is compared against the **strongest possible version** of the Bo2 baseline, and still delivers improvements.
>
> ------
>
> ## **[Q1 & Q2] Future Directions**
>
> We thank the reviewer for the insightful questions regarding future directions of RFG.
>
> **Q1. Using a single dLLM with and without prompt.**
> We see this as a promising direction and add discussion in the paper. However, we note that the **theoretical decomposition for RFG does not hold** for this setup, so similar performance is not guaranteed.
>
> **Q2. Using implicit rewards for RL training of dLLMs.**
> Our framework naturally provides a step-wise reward signal that could be used for RL training. While our focus is test-time scaling, this is an exciting direction that we now discuss in the revision.

---

> ### Author Response · Authors · 2025-11-26
>
> ## **Summary**
>
> We have clarified that:
>
> - RFG is *convenient*: it uses publicly released checkpoints *off-the-shelf* and requires *minimal tuning*.
> - Our reward formulation and derivation is *rigorous and self-contained*, and we have *added additional intuition for accessibility*.
> - *Performance gains of RFG are substantial* for a training-free method. The ensemble baseline is an *important compute-matched control*, and Bo2 underperforms RFG.
> - The reviewer’s proposed future directions are promising, and we have added discussion accordingly.
>
> We hope these clarifications address the reviewer’s concerns and welcome further discussion.

---

### Official Review · Reviewer_z2d3 · 2025-11-02

**Soundness:** 3
**Presentation:** 3
**Contribution:** 2
**Rating:** 4
**Confidence:** 4

**Summary:**

This paper aims to address a key challenge in diffusion large language models (dLLMs): unlike autoregressive (AR) LLMs, dLLMs generate text in an "any-order" fashion, resulting in intermediate states of partially masked sentences that make it difficult to train explicit process reward models (PRMs) for step-wise reasoning guidance. To solve this, the authors propose Reward-Free Guidance (RFG), a test-time framework that avoids explicit PRM training by parameterizing rewards as the log-likelihood ratio between two dLLMs: a "policy model" ($p_{\theta}$, typically a post-trained dLLM via RL or SFT, e.g., d1-LLaDA, DiffuCoder) and a "reference model" ($p_\text{ref}$, a base pre-trained dLLM, e.g., LLaDA-Base).

In experiments, RFG is evaluated on mathematical reasoning (GSM8K, MATH-500) and code generation (HumanEval, MBPP) across multiple dLLM families (LLaDA, Dream).
The authors report consistent accuracy gains (up to 9.2% for DiffuCoder on HumanEval) over two baselines: the original post-trained $p_{\theta}$ and a naive ensemble of $p_{\theta}$ and $p_\text{ref}$.

Overall, the work identifies a relevant problem for dLLMs but lacks sufficient technical depth and novelty. Its performance gains heavily rely on the pre-optimized quality of $p_{\theta}$ (rather than RFG’s intrinsic guidance mechanism) and fails to address the core challenge of dLLM-specific intermediate state reasoning.

**Strengths:**

1. This paper targets a high-priority gap in dLLM inference—masked intermediate states (e.g., "[MASK] + 2 = 5") that invalidate AR-LLM process reward model (PRM) methods. Such a challenge directly limits dLLMs’ practical reasoning scalability, making the work’s focus timely and impactful.

2. The introduced training-free guidance framework bypasses explicit PRM training. This avoids the "reward annotation bottleneck" of RL-based dLLM methods and delivers consistent accuracy gains (up to 9.2% for DiffuCoder on HumanEval) across diverse tasks.

**Weaknesses:**

1. Lacks dLLM-specific innovation—essentially repurposes Classifier-Free Guidance by swapping conditional/unconditional models. RFG’s log-likelihood ratio guidance mirrors classifier-free guidance, a generic diffusion technique, with no dLLM-specific adaptations (e.g., mask-order-aware weighting). The "reward-free" claim is misleading, as it relies on  implicit RL/SFT rewards.

2. Gains are ambiguously attributed (e.g., +0.5% on GSM8K for LLaDA 1.5 may stem from $p_\theta$ quality, not RFG). Further, RFG lacks "scaling" behavior—performance does not improve with increased test-time computation (e.g., more sampling steps), violating the essence of test-time scaling.

**Questions:**

1. How is the "naive ensemble" logit average computed (per-token vs. sequence-level)?

2. How about the performance gains with the computation budget scaling, based on RFG?

---

> ### Author Response · Authors · 2025-11-26
>
> We thank the reviewer for their review and for acknowledging our work as "timely and impactful" and our method achieves "consistent gains across diverse tasks". We also appreciate their feedback and address each concern below.
>
> ------
>
> ## **[W1 first half] dLLM-Specific Innovation**
>
> The reviewer states that RFG lacks dLLM-specific innovation over CFG. We make expilict comparison with CFG in the paper. While the *mathematical form* is similar, we highlight RFG's innovation beyond CFG as follows.
>
> RFG's innovation is the conceptual leap in *what* is being guided and *how* it solves a new, dLLM-specific problem.
>
> 1. **A Novel, dLLM-Specific Problem:**
>
>    CFG guides on a simple *class label*.
>
>    **RFG addresses a far more complex, dLLM-specific challenge**: guiding a generative process toward *correct, multi-step reasoning* when the intermediate states are partially masked sentences.
>
>    This is a problem **CFG does *not* address**, as CFG offers no notion of process reward shaping.
>
> 2. **A Novel Theoretical Justification:**
>
>    Our key innovation is providing a novel theoretical justification (Prop. 3.1) showing how a trajectory-level reward can be **decomposed into a valid PRM for each diffusion step**.
>
>    This allows us to apply a step-wise guidance signal that is specifically tailored to the diffusion process, which is ***not* covered by CFG** or any prior dLLM work.
>
> In short, leveraging the mathmetical form of guidance in a *new, non-trivial domain* (dLLM reasoning) with a *new theoretical grounding* (decomposed process reward) that *addresses non-trivial domain-specific challenge* (process reward for partially masked sequence) is a significant and dLLM-specific innovation.
>
> ---
>
> ## **[W1 second half] Reward-free Definition**
>
> The reviewer claims that "reward-free" is misleading because RFG still relies on the implicit rewards from the SFT/RL training of the policy model.
>
> Our "reward-free" claim is a direct analogy to *"classifier-free"* in CFG.
>
> CFG is named "classifier-free" because it removes the need for an *explicit, separately-trained classifier network*. It still **relies on the *implicit* classifier score** derived from the difference between the conditional and unconditional score estimates.
>
> RFG is named "reward-free" for the same reason: it **removes the need for an *explicit, separately-trained reward model***. This is our core contribution.
>
> By this same logic, we believe our naming is consistent and appropriate. The key benefit is that RFG is a completely **training-free** framework that works at test time.
>
> ------
>
> ## **[W2 first half] Attributing Performance Gains**
>
> The reveiewer suspects that gains may stem from the quality of optimized model $p_{\theta}$, using LLaDA 1.5 on GSM8K as an example.
>
> We thank the reviewer for pointing to this specific example, which we believe proves the opposite.
>
> Our experimental design **explicitly controls for the quality of $p_{\theta}$** by including the **"Original"** baseline, which is *just* $p_{\theta}$'s performance. **RFG's gain is the *delta* on top of that**.
>
> The reviewer's chosen example, LLaDA 1.5 on GSM8K, is an RL-optimized $p_{\theta}$ of higher quality than a SFT $p_{\theta}$ (LLaDA Instruct). However, RFG provides a *larger* gain on a *weaker* $p_{\theta}$ (LLaDA Instruct). This shows that the performance gain brought by RFG does **not stem from the $p_{\theta}$ quality.**
>
> In fact, the gain is attributed to RFG's own principled guidance, which **delivers consistent gains *on top of* the original model $p_{\theta}$**, not just a proxy for its quality.
>
> We've provide additional results where we conduct extensive experiments to validate RFG's effectiveness under different diffusion settings (see Table 3- 10). We hope the additional results and the clarification above could address the reviewer's concern and demonstrate the effectiveness of RFG stems from our principled formulation.
>
> ---
>
> ## **[W2 second half & Q2] Test-time Scaling Explanation**
>
> The reviewer asks for explanation why RFG is a test-time scaling method.
>
> We clarify that RFG represents a form of efficient scaling: **it utilizes increased inference compute** (via the additional reference model) **to achieve performance gains that would otherwise require expensive training**. While it is a fixed cost increase (2x) rather than an iterative search, it fits the definition of trading inference flops for accuracy.
>
> ---
>
> ## **[Q1] Details of Naive Ensemble Baseline**
>
> The reviewer asks if the logits for naive ensemble baseline is averaged per-token or sequence-level.
>
> The average is computed **per-token, at every denoising step**. As detailed in Section 4.1, at each step, we compute the logit from both $p_{\theta}$ and $p_{ref}$ and then average them.
>
> This was specifically designed as an **identical-compute-budget baseline** to ensure our gains from RFG were not merely from increased computation.

---

> ### Author Response · Authors · 2025-11-26
>
> ## **RFG Is Effective Across Different Diffusion Settings** (Additional Results for [W2 first half] Attributing Performance Gains)
> ### **Different Denoising Steps**
> We've added experiments across denoising steps of {128, 256, 512}, with results shown in Table 3-8. RFG **achieves consistent improvement over baseline on all benchmarks and for all models**.
> ### #steps=128
> Table 3: Performance on GSM8K and MATH-500, with the number of denoising step 128. Best in **bold**.
> ||GSM8K|GSM8K|GSM8K|MATH-500|MATH-500|MATH-500|
> |:-|:-:|:-:|:-:|:-:|:-:|:-:|
> |**Model**|Original|Ensemble|RFG|Original|Ensemble|RFG|
> |LLaDA 8B Instruct|76.3|39.3|**77.2**|32.4|22.8|**34.2**|
> |Dream 7B Instruct|67.9|27.7|**74.2**|32.2|11.6|**33.2**|
> |d1-LLaDA|79.3|42.4|**79.8**|34.0|25.2|**37.4**|
> |LLaDA 1.5|76.3|45.3|**78.0**|32.0|23.0|**34.0**|
>
> Table 4: Performance on HumanEval and MBPP, with the number of denoising step 128. Best in **bold**.
> ||HumanEval|HumanEval|HumanEval|MBPP|MBPP|MBPP|
> |:-|:-:|:-:|:-:|:-:|:-:|:-:|
> |**Model**|Original|Ensemble|RFG|Original|Ensemble|RFG|
> |LLaDA 8B Instruct|32.3|18.3|**33.5**|51.0|52.5|**54.1**|
> |Dream 7B Instruct|52.4|48.2|**60.4**|65.0|65.8|**68.1**|
> |diffuCoder|70.1|64.6|**73.8**|72.4|70.0|**73.9**|
>
> ### #steps=256
> Table 5: Performance on GSM8K and MATH-500, with the number of denoising step 256. Best in **bold**.
> ||GSM8K|GSM8K|GSM8K|MATH-500|MATH-500|MATH-500|
> |:-|:-:|:-:|:-:|:-:|:-:|:-:|
> |**Model**|Original|Ensemble|RFG|Original|Ensemble|RFG|
> |LLaDA 8B Instruct|79.8|68.0|**81.3**|38.4|33.6|**39.6**|
> |Dream 7B Instruct|80.9|68.3|**82.1**|43.6|30.6|**46.4**|
> |d1-LLaDA|82.5|68.5|**84.7**|38.8|34.8|**41.6**|
> |LLaDA 1.5|81.6|70.9|**82.1**|38.6|34.2|**39.6**|
>
> Table 6: Performance on HumanEval and MBPP, with the number of denoising step 256. Best in **bold**.
> ||HumanEval|HumanEval|HumanEval|HumanEval|MBPP|MBPP|
> |:-|:-:|:-:|:-:|:-:|:-:|:-:|
> |**Model**|Original|Ensemble|RFG|Original|Ensemble|RFG|
> |LLaDA 8B Instruct|39.6|37.2|**40.9**|54.9|51.4|**55.6**|
> |Dream 7B Instruct|61.0|59.1|**66.5**|64.2|67.3|**68.5**|
> |diffuCoder|69.5|72.6|**76.2**|72.4|72.8|**73.5**|
>
> ### #steps=512
>
> Table 7: Performance on GSM8K and MATH-500, with the number of denoising step 512. Best in **bold**.
> ||GSM8K|GSM8K|GSM8K|MATH-500|MATH-500|MATH-500|
> |:-|:-:|:-:|:-:|:-:|:-:|:-:|
> |**Model**|Original|Ensemble|RFG|Original|Ensemble|RFG|
> |LLaDA 8B Instruct|84.1|80.1|**84.7**|42.6|37.6|**43.6**|
> |Dream 7B Instruct|84.8|84.0|**85.1**|50.4|51.2|**52.6**|
> |d1-LLaDA|82.8|80.1|**84.7**|41.6|38.8|**45.4**|
> |LLaDA 1.5|84.7|80.5|**85.4**|42.4|40.6|**44.4**|
>
> Table 8: Performance on HumanEval and MBPP, with the number of denoising step 512. Best in **bold**.
> ||HumanEval|HumanEval|HumanEval|MBPP|MBPP|MBPP|
> |:-|:-:|:-:|:-:|:-:|:-:|:-:|
> |**Model**|Original|Ensemble|RFG|Original|Ensemble|RFG|
> |LLaDA 8B Instruct|45.1|40.9|**47.6**|49.8|49.8|**52.9**|
> |Dream 7B Instruct|62.2|58.5|**65.2**|64.6|69.6|**70.4**|
> |diffuCoder|69.5|73.2|**78.7**|72.8|70.8|**74.3**|
>
>
> ### **Multi-token Prediction**
> We've added experiments for all models and all benchmarks under multi-token prediction setup by doubling the number of token per denoising step in Table 1 and 2, with results shown in Table 9 and 10. The results demonstrates that RFG achieves consistent improvement on multi-token prediction setup.
>
> Table 9: Performance on GSM8K and MATH-500, doubling the number of token per denoising step in Table 1.
> ||GSM8K|GSM8K|GSM8K|GSM8K|MATH-500|MATH-500|MATH-500|MATH-500|
> |:-|:-:|:-:|:-:|:-:|:-:|:-:|:-:|:-:|
> |**Model**|Original|Ensemble|RFG|Rel. Gain|Original|Ensemble|RFG|Rel. Gain|
> |LLaDA 8B Instruct|78.6|64.3|**80.7**|**+2.1**|34.2|29.4|**38.0**|**+3.8**|
> |Dream 7B Instruct|68.5|58.6|**73.6**|**+5.1**|35.4|27.4|**37.6**|**+2.2**|
> |d1-LLaDA|79.8|67.3|**82.0**|**+2.2**|36.6|29.4|**39.2**|**+2.6**|
> |LLaDA 1.5|81.5|70.2|**82.1**|**+0.6**|38.6|37.2|**41.4**|**+2.8**|
>
> Table 10: Performance on HumanEval and MBPP, doubling the number of token per denoising step in Table 2.
> ||HumanEval|HumanEval|HumanEval|HumanEval|MBPP|MBPP|MBPP|MBPP|
> |:-|:-:|:-:|:-:|:-:|:-:|:-:|:-:|:-:|
> |**Model**|Original|Ensemble|RFG|Rel. Gain|Original|Ensemble|RFG|Rel. Gain|
> |LLaDA 8B Instruct|37.2|32.9|**38.4**|**+1.2**|43.6|39.7|**44.4**|**+0.8**|
> |Dream 7B Instruct|37.8|39.0|**42.7**|**+4.9**|48.2|52.1|**55.3**|**+7.1**|
> |diffuCoder|57.3|51.2|**60.4**|**+3.1**|61.1|36.6|**63.0**|**+1.9**|

---

> ### Author Response · Authors · 2025-11-26
>
> ## **Summary**
>
> We provided clarifications that:
>
> - RFG has dLLM-specific innovation over CFG: it contributes a *new theoretical grounding* and *dLLM-specific mechanisms* for partially-masked and any-order trajectories.
> - Reward-free means no explicit, seperately trained reward model.
> - The consistent gains of RFG should not be attributed to optimized model $p_{\theta}$ quality. The gains stem from our principled formulation.
> - RFG is a test-time scaling method that does scale with compute.
>
> We hope these clarifications address the reviewer’s concerns and are happy to engage in further discussion.

---

### Official Review · Reviewer_Vyfw · 2025-11-02

**Soundness:** 2
**Presentation:** 3
**Contribution:** 2
**Rating:** 4
**Confidence:** 4

**Summary:**

This paper proposes RFG (reward-free guidance) for diffusion LLM test-time sampling to achieve better reasoning performance. They prove that trajectory-level rewards can be decomposed into per-denoising-step rewards, then use the log-likelihood ratio between an enhanced model (RL-trained or instruction-tuned) and a reference base model to guide the sampling process at test time. The method shows performance gains on math and coding tasks with qualitative examples demonstrating how RFG achieves better results, outperforming both the original enhanced models and naive ensemble baselines with matched compute.

**Strengths:**

1. this paper provides a clean theoretical justification showing how trajectory-level rewards can be decomposed into step-wise process rewards through log-likelihood ratios for diffusion LLMs. it tackles the difficulty that obtaining a process reward model for diffusion LLM as dLLM will generate partially denoised sequences and it would be hard to train such PRM.
2. This paper shows their proposed sampling method RFG achieves consistent gain across pairs of dLLMs on mathematics and coding tasks and it is also training-free.

**Weaknesses:**

1. lack of ablation studies (with only one ablation presented). More analysis would strength the work on how RFG actually changes the decoding process, for example how it affects the denoising order (a key feature of dLLMs' any-order generation). The paper only tests in the setting: "the number of generation steps is set equal to the generation length",  but the core advantage of dLLMs is parallel multi-token generation. The paper provides no ablation on how block size/number of denoising steps affects RFG performance, and no exploration of whether decoding more than 2 tokens per step changes the effectiveness of guidance. this limits its practical value for future dLLMs.

    Since RFG requires running two full dLLM forward passes at every denoising step, which is very expensive for full-attention based dLLMs like LLaDA and Dream and the fact they only tested on sampling 1 token per denoising step makes this method less practical, if RFG improves on multi-token prediction setup that is more promising.

6. The paper motivates RFG by arguing that training explicit PRMs for dLLMs is challenging due to incomplete intermediate states. However, RFG is more like a guided sampling method that reweights each denoising step, rather than a search-based approach that PRMs typically enable in autoregressive LLM contexts (e.g., beam search, tree search with PRM scoring)
1. how would RFG compare against simply using lower temperature sampling on the enhanced model alone, as it would sharpen the probability distribution and potentially achieving similar effects to RFG's guidance by making the model more confident in its existing preferences.

2. The theoretical derivation assumes the enhanced model is trained via RL/preference optimization, However, instruction-tuned models represent a major portion of their experiments (LLaDA-Instruct, Dream-Instruct) and show substantial gains. The paper lacks explanation for why SFT models work as their main theoretical contribution doesn't cover a significant portion of their empirical results.

**Questions:**

see weakness above.
1. Does RFG also work for block-diffusion sampling for models like SDAR or trado?

---

> ### Author Response · Authors · 2025-11-26
>
> We thank the reviewer for their review and for acknowledging the clean theoretical justification and consistently strong empirical results of our method, as well as its training-free advantage. We also appreciate their feedback and address each concern below.
>
> ---
>
> ## **[W1] Additional Ablation Studies**
>
> The reviewer suggests more ablation studies on varying generation steps and parallel multi-token generation setting.
>
> We appreciate the suggestion and conduct more experiments, with results shown in Table 3-10.

---

> ### Author Response · Authors · 2025-11-26
>
> ## **(cont) [W1] Additional Ablation Studies**
>
> ### **RFG consistently improves performance across different numbers of denoising steps.**
>
> We have added experiments for all models on all benchmarks across denoising steps of {128, 256, 512}, with results shown in Table 3- 8.
>
> The results demonstrate that RFG **achieves consistent improvement over baseline across different number of denoising steps on all benchmarks and for all models**, highlighting its effectiveness and robustness across different settings.
>
> ### #steps=128
> Table 3: Performance on GSM8K and MATH-500, with the number of denoising step 128. Best in **bold**.
> ||GSM8K|GSM8K|GSM8K|MATH-500|MATH-500|MATH-500|
> |:-|:-:|:-:|:-:|:-:|:-:|:-:|
> |**Model**|Original|Ensemble|RFG|Original|Ensemble|RFG|
> |LLaDA 8B Instruct|76.3|39.3|**77.2**|32.4|22.8|**34.2**|
> |Dream 7B Instruct|67.9|27.7|**74.2**|32.2|11.6|**33.2**|
> |d1-LLaDA|79.3|42.4|**79.8**|34.0|25.2|**37.4**|
> |LLaDA 1.5|76.3|45.3|**78.0**|32.0|23.0|**34.0**|
>
> Table 4: Performance on HumanEval and MBPP, with the number of denoising step 128. Best in **bold**.
> ||HumanEval|HumanEval|HumanEval|MBPP|MBPP|MBPP|
> |:-|:-:|:-:|:-:|:-:|:-:|:-:|
> |**Model**|Original|Ensemble|RFG|Original|Ensemble|RFG|
> |LLaDA 8B Instruct|32.3|18.3|**33.5**|51.0|52.5|**54.1**|
> |Dream 7B Instruct|52.4|48.2|**60.4**|65.0|65.8|**68.1**|
> |diffuCoder|70.1|64.6|**73.8**|72.4|70.0|**73.9**|
>
> ### #steps=256
> Table 5: Performance on GSM8K and MATH-500, with the number of denoising step 256. Best in **bold**.
> ||GSM8K|GSM8K|GSM8K|MATH-500|MATH-500|MATH-500|
> |:-|:-:|:-:|:-:|:-:|:-:|:-:|
> |**Model**|Original|Ensemble|RFG|Original|Ensemble|RFG|
> |LLaDA 8B Instruct|79.8|68.0|**81.3**|38.4|33.6|**39.6**|
> |Dream 7B Instruct|80.9|68.3|**82.1**|43.6|30.6|**46.4**|
> |d1-LLaDA|82.5|68.5|**84.7**|38.8|34.8|**41.6**|
> |LLaDA 1.5|81.6|70.9|**82.1**|38.6|34.2|**39.6**|
>
> Table 6: Performance on HumanEval and MBPP, with the number of denoising step 256. Best in **bold**.
> ||HumanEval|HumanEval|HumanEval|HumanEval|MBPP|MBPP|
> |:-|:-:|:-:|:-:|:-:|:-:|:-:|
> |**Model**|Original|Ensemble|RFG|Original|Ensemble|RFG|
> |LLaDA 8B Instruct|39.6|37.2|**40.9**|54.9|51.4|**55.6**|
> |Dream 7B Instruct|61.0|59.1|**66.5**|64.2|67.3|**68.5**|
> |diffuCoder|69.5|72.6|**76.2**|72.4|72.8|**73.5**|
>
> ### #steps=512
>
> Table 7: Performance on GSM8K and MATH-500, with the number of denoising step 512. Best in **bold**.
> ||GSM8K|GSM8K|GSM8K|MATH-500|MATH-500|MATH-500|
> |:-|:-:|:-:|:-:|:-:|:-:|:-:|
> |**Model**|Original|Ensemble|RFG|Original|Ensemble|RFG|
> |LLaDA 8B Instruct|84.1|80.1|**84.7**|42.6|37.6|**43.6**|
> |Dream 7B Instruct|84.8|84.0|**85.1**|50.4|51.2|**52.6**|
> |d1-LLaDA|82.8|80.1|**84.7**|41.6|38.8|**45.4**|
> |LLaDA 1.5|84.7|80.5|**85.4**|42.4|40.6|**44.4**|
>
> Table 8: Performance on HumanEval and MBPP, with the number of denoising step 512. Best in **bold**.
> ||HumanEval|HumanEval|HumanEval|MBPP|MBPP|MBPP|
> |:-|:-:|:-:|:-:|:-:|:-:|:-:|
> |**Model**|Original|Ensemble|RFG|Original|Ensemble|RFG|
> |LLaDA 8B Instruct|45.1|40.9|**47.6**|49.8|49.8|**52.9**|
> |Dream 7B Instruct|62.2|58.5|**65.2**|64.6|69.6|**70.4**|
> |diffuCoder|69.5|73.2|**78.7**|72.8|70.8|**74.3**|
>
> ### **RFG improves performance on multi-token prediction setup**
>
> We have added experiments for all models and all benchmarks under multi-token prediction setup by doubling the number of token per denoising step in Table 1 and 2, with results shown in Table 9 and 10.
>
> The results demonstrates that RFG achieves consistent improvement on multi-token prediction setup, strengthening its effectiveness and applicability.
>
> Table 9: Performance on GSM8K and MATH-500, doubling the number of token per denoising step in Table 1.
> ||GSM8K|GSM8K|GSM8K|GSM8K|MATH-500|MATH-500|MATH-500|MATH-500|
> |:-|:-:|:-:|:-:|:-:|:-:|:-:|:-:|:-:|
> |**Model**|Original|Ensemble|RFG|Rel. Gain|Original|Ensemble|RFG|Rel. Gain|
> |LLaDA 8B Instruct|78.6|64.3|**80.7**|**+2.1**|34.2|29.4|**38.0**|**+3.8**|
> |Dream 7B Instruct|68.5|58.6|**73.6**|**+5.1**|35.4|27.4|**37.6**|**+2.2**|
> |d1-LLaDA|79.8|67.3|**82.0**|**+2.2**|36.6|29.4|**39.2**|**+2.6**|
> |LLaDA 1.5|81.5|70.2|**82.1**|**+0.6**|38.6|37.2|**41.4**|**+2.8**|
>
> Table 10: Performance on HumanEval and MBPP, doubling the number of token per denoising step in Table 2.
> ||HumanEval|HumanEval|HumanEval|HumanEval|MBPP|MBPP|MBPP|MBPP|
> |:-|:-:|:-:|:-:|:-:|:-:|:-:|:-:|:-:|
> |**Model**|Original|Ensemble|RFG|Rel. Gain|Original|Ensemble|RFG|Rel. Gain|
> |LLaDA 8B Instruct|37.2|32.9|**38.4**|**+1.2**|43.6|39.7|**44.4**|**+0.8**|
> |Dream 7B Instruct|37.8|39.0|**42.7**|**+4.9**|48.2|52.1|**55.3**|**+7.1**|
> |diffuCoder|57.3|51.2|**60.4**|**+3.1**|61.1|36.6|**63.0**|**+1.9**|
>
> We thank the reviewer for the suggestion and have updated the paper to include these results.

---

> ### Author Response · Authors · 2025-11-26
>
> ## **[W2] Connection to PRM**
>
> The reviewer notes that RFG is a guided sampling method, not a search method which is how Process Reward Models (PRMs) are typically used in autoregressive (AR) LLMs.
>
> Our paper decouples two distinct concepts:
>
> 1. The **PRM signal:** This is simply a reward signal, $r^t$, that assigns a quality score to an intermediate step $t$ of a generation.
> 2. The **Algorithm that uses the PRM signal:** This is *how* the reward signal is used to improve generation (e.g., search, filtering, or guidance).
>
> Our paper makes two key claims on these two concepts:
>
> 1. **RFG implicitly defines a PRM signal:** Our theoretical derivation (Prop. 3.1) and resulting step-wise reward in Eq. 2 **is by definition a PRM signal**. It assigns a score to each intermediate denoising transition, fulfilling the role of a process reward.
>
> 2. **RFG uses this PRM signal for guidance:** RFG applies this PRM signal in a different but equally valid way: as a guidance signal rather than a search signal. Such guidance signal reweight the sampling distribution at each step, as shown in Eq. 3.
>
> This distinction clarifies our motivation.
>
> When we state that "training explicit PRMs for dLLMs is challenging", we are **highlighting the difficulty of obtaining *any* reliable step-wise signal for dLLMs**.
>
> Our paper's contribution is twofold:
>
> 1. We show **how to *implicitly define* a PRM without training**.
> 2. We propose a novel way to ***apply* this PRM signal** (via guided sampling) that is a natural and effective fit for the diffusion process.
>
> ---
>
> ## **[W3] Comparison to Baseline using Low-Temperature Sampling**
>
> The reviewer insightfully asks whether the gains from RFG could be replicated by simply lowering the sampling temperature of the enhanced model alone, which would also sharpen the distribution.
>
> Our "Original" baselines of enhanced model in Tables 1 and 2 are **already run using the optimized low-temperature settings** recommended in their respective original papers. This ensures we are comparing RFG against the strongest possible version of the baseline model.
>
> Specifically, for all LLaDA-family models, we follow their official setup and use a temperature of 0 (i.e., deterministic greedy decoding). For all Dream-family models, we follow their setup and use a temperature of 0.1 (setting temperature to 0 for Dream models causes a performance collapse, as the model will exhibit repetition behavior and frequetly generate a whole sequence with eos/pad tokens.)
>
> Therefore, the consistent performance gains from RFG shown in Tables 1 and 2 are **improvements *on top of* an already optimized, low-temperature baseline**.
>
> This demonstrates that RFG is performing a fundamentally different and more powerful operation than simple temperature tuning.
>
> ---
>
> ## **[W4] Explanation on Empirical Gains for SFT Models**
>
> The reviewer notes that our theory is derived from an RL setting and our experiments on SFT models also show substantial gains, and suggests for a clearer explanation.
>
> We thank the reviewer for pushing us to clarify this important point.
>
> First, we'd like to note that our **theoretical derivation aligns perfectly with the empirical improvements on RL models**. Additionally, we noticed that the **RL-optimized dLLMs is relatively scarce** in dLLM literature, as directly applying exisiting RL method like GRPO and DPO to dLLMs is not feasible. Given this, we consider applying RFG to SFT models that are easier to obtain, and we're glad to see significant gains as well.
>
> We agree that explaining *why* RFG works so well for SFT is crucial and provide the explanation as follows.
>
> RFG formulation can be rewritten as $\log \pi_{RFG} = \log~\pi_\theta + w(\log~\pi_\theta - \log~\pi_\text{ref})$.
>
> We can interpret each term as follows:
>
> - $\log~\pi_{ref}$ is the distribution of the base model, which predicts "general text."
> - $\log~\pi_{\theta}$ is the distribution of the SFT model, which is fine-tuned to predict "instruction-following text."
> - The **difference vector** $(\log~\pi_{\theta} - \log~\pi_{ref})$ therefore represents the *optimization direction* that SFT applied. It is a vector that points *away* from "general text" and *towards* "instruction-following" text.
>
> RFG's formulation, therefore, ***steers* the SFT model's distribution ($\log~\pi_{\theta}$) further along this exact optimization direction, effectively amplifying the "knowledge" that the SFT model learned**. This explains why RFG works well for SFT models, pushing them to be even better at the task they were fine-tuned for.
>
> The fact that RFG provides significant gains for both RL-tuned and SFT-tuned models is a key finding of our paper, highlighting the robustness and generality of our approach.
>
> We thank the reviewer for the insightful suggestion and have updated the paper to include the explanation.

---

> ### Author Response · Authors · 2025-11-26
>
> ## **[Q1] RFG for Block Diffusion Sampling**
>
> We thank the reviewer for bringing up the excellent concurrent work SDAR[1] and TraDo[2]. We've cited both of them and discussed how to apply RFG to them.
>
> [1] Cheng, Shuang, et al. "SDAR: A Synergistic Diffusion-AutoRegression Paradigm for Scalable Sequence Generation." *arXiv preprint arXiv:2510.06303* (2025).
>
> [2] Wang, Yinjie, et al. "Revolutionizing reinforcement learning framework for diffusion large language models." *arXiv preprint arXiv:2509.06949* (2025).
>
> ---
>
> ## **Summary**
>
> We have
>
> - conducted additional experiments to demonsrate RFG's effectiveness across different denoising settings (Tables 3-10).
> - clarified RFG's connection to PRM and our motivation.
> - provided baseline results under low-temperature sampling.
> - explained why RFG works for SFT models.
>
> We hope these clarifications address the reviewer’s concerns and are happy to engage in further discussion.

---

### Official Review · Reviewer_o6q9 · 2025-11-08

**Soundness:** 2
**Presentation:** 2
**Contribution:** 2
**Rating:** 4
**Confidence:** 4

**Summary:**

The paper introduces RFG (reward free guidance), a novel method for guiding the generation of diffusion language models at test time. RFG as a test time method makes use of information from a reference language model to guide the diffusion process of dLLM. The paper showcases improvement over a large class of dLLMs (instructioned tuned, RL'ed) showcasing the power of the new approach.

**Strengths:**

The paper provides an interesting way to combine reward-free guidance, that is, not training an explicit reward for guiding test time generation for dLLM. They introduce a somewhat novel and sensible way to leverage logit information from a reference model. Their experimental result is also showing relatively solid improvements.

**Weaknesses:**

I think there are a few practical constraints of the proposed method that limit its applicability, and the way ablations and comparisons are conducted might also warrant additional scrutiny and improvement. I will detail these points in the question section below.

**Questions:**

=== **test time cost measurement of using a reference model** ===

I think it's quite important to calibrate for the actual cost at test time when making the comparison in e.g. Table 2. The algorithm proposed need to query reference model every iteration, which is extremely costly, given that dLLM generations typically denoise for a relatively large number of steps (less than generation length though). I think this is an important axis to ablate - in additional to the strength $w$, the number of query made to the reference model is also a big factor controlling the cost of the test time algorithm. If the number of query is such that it leads to a cost much higher than alternative baselines, the gains are much less surprising.

Note the Ensemble baseline essentially runs the original diffusion model multiple times, but if the reference model is of a bigger size, the cost of Ensemble is much smaller than RFG. What if we run Ensemble the same order of cost magnitude as RFG and compare the performance difference?

It is helpful to have another axis for RFG that controls the cost of the test time budget, and examine how performance fares as this varies.

=== **tokenizer** ===

It seems that to apply RFG we require the same tokenizer for the dLLM and reference model because we require their logits to be additive. This might form a constraint because we'd expect tokenizers to be more flexible across model family and this will naively limit the applicability. I think it's also useful to work out a solution for such cases.

=== **RFG against other baselines** ===

I think other baselines are worth comparing to such as training actual outcome reward for dLLM, or making use of existing outcome reward for those tasks and use the best-of-n test time budget to see whether there is a performance difference. RFG comparison directly skips through the whole outcome reward baselines and test time algorithm, which I think are very important to compare against. Such methods are also arguably cheaper than RFG because outcome rewards need to be queried fewer times than ref model.

=== **Reference model** ===

RFG is quite generic and arguably applicable to a big class of ref model, but I think it's useful to ablate how the strengths of the ref model impacts generation quality and how sensitive is the performance to the $w$ coefficient.

---

> ### Author Response · Authors · 2025-11-26
>
> We thank the reviewer for their review and for acknowledging the novelty and solid empirical results of our method. We also appreciate their feedback and address each concern below.
>
> ---
>
> ## **[Q1] Test-Time Cost and Compute-Budget-Matched Comparison**
>
> The reviewer is concerned about the test time cost of RFG and asks for a comparison against baselines with an identical compute budget.
>
> We'd like to clarify that the ensemble baseline is **specifically designed to be an identical-compute-budget baseline for RFG**.
>
> **Reference Model Selection:**
>
> - The reference model in our experiment is the base version of the post-trained dLLMs and they are of the **same size** (e.g., LLaDA-8B-Base as reference model for LLaDA-8B-Instruct).
>
> - This choice is convinient as it both ensures same tokenizer so that logits are addable and does not need any training since both checkpoints of base and post-trained models are publicly available and can be used off-the-shelf.
>
> **Identical Computational Cost**:
>
> - Ensemble baseline requires one forward pass through the base (reference) model and one forward pass through the post-trained (policy) model, and it then computes the final logits by averaging the logits from both.
>
> - RFG uses the *same compute budget*: one forward pass through the base (reference) model and one forward pass through the post-trained (policy) model, as shown in Algorithm 1.
>
> Therefore, the **ensemble baseline and RFG have the exact same computational cost** in all experiments in Tables 1 and 2.
>
> The fact that RFG **consistently and significantly outperforms this perfectly cost-matched ensemble baseline** proves that the performance gain comes from our principled formulation, not merely from incorporating the reference model's parameters or more compute.
>
> ---
>
> ## **[Q2] Tokenizer**
>
> The reviewer is concerned if RFG's applicability will be limited by the oberservation that the policy and reference models must share the same tokenizer.
>
> We'd like to clarify that this is *not* a practical constraint for the intended use case of RFG, and that RFG is broadly applicable.
>
> **Reference Model Selection**:
>
> - The reference model in RFG is the base version where the post-trained dLLMs are trained from (e.g., LLaDA Base as reference model for d1-LLaDA and Dream Base for diffuCoder).
> - This choice inherently ensures same tokenizer so that logits are addable.
> - We **don't expect tokenizers to be flexible across model family** and we use different reference models for different model families, ensuring broad applicability.
>
> **Broadly Applicable in Practice**:
>
> - The practical use case for RFG is to enhance an existing post-trained model using its own original pre-trained base model as the reference.
> - Since the checkpoints of both the base and post-trained models are typically publicly available off-the-shelf, our method is **completely training-free**, warranting broad applicability.
>
> **Experimental Results Demonstrate Broad Applicability**:
>
> - As our experiments show (Tables 1 & 2), RFG is already broadly applicable, showing **consistent gains** across different model families (LLaDA, Dream), post-training methods (SFT, RL), and tasks (math, code).
>
> In short, RFG selects reference model in a way that inherentlty satisfy the logits addablility and doesn't require tokenizer to be flexible across model family. Empirical results show consistent improvement across different model families, post-trained methods, and tasks, which demonstrate RFG's broad applicability.
>
> ---
>
> ## **[Q3] Additional Best-of-N Baselines**
>
> The reviewer asks for a comparison with Best-of-N (BoN) sampling using outcome reward, suggesting such methods might be cheaper.
>
> We'd like to clarify that BoN is not cheaper than RFG, and RFG outperform BoN under same compute budget.
>
> **BoN is Not Cheaper:**
>
> - At inference, BoN requires N forward passes of the model.
>
> - RFG requires only two (note that the reference model and policy model are of the same size and same architecture, so each forward pass has almost identical computational cost).
>
> - Therefore, BoN is more expensive than RFG when $N>2$, and is of same compute budget as RFG when $N=2$.
>
> **BoN Underperforms RFG**:
>
> - We conduct experiments for BoN ($N=2$), which uses same compute budget as RFG.
> - BoN ($N=2$) is effectively identical to the original, post-trained model’s performance, i.e., **BoN underperforms RFG** under same compute budget.

---

> ### Author Response · Authors · 2025-11-26
>
> ## **[Q4] Ablations**
>
> The reviewer asks for two ablations: on the impact of the reference model's strength, and on the sensitivity to the guidance strength $w$.
>
> ### **Ablation on Strength of Reference Model**
>
> This is an interesting research question. Our choice of reference model was motivated by practicality (training-free and off-the-shelf applicability), where we directly take *existing checkpoints of the base model* as reference model.
>
> Developing reference model of different strength may **require additional training** (e.g., diliberately train a weaker version of the post-trained dLLMs), which is beyond our focus and we leave it as a future direction.
>
> ### **Ablation on Sensitivity to Guidance Strength $w$**
>
> We've conducted such ablation study with results in **Fig. 3 (Fig. 2 in updated version) Sec. 4.3**.
>
> RFG exhibits a wide plateau of strong performance across a broad range of $w$ values.
>
> This robustness is a key practical advantage of RFG.
>
> ---
>
> ## **Summary**
>
> We've clarified that:
>
> - The baselines use *same compute budget* as RFG.
> - RFG does *not* require same tokienzer for different model families and is applicable to different model families.
> - BoN is more expensive than RFG when $N>2$, and BoN underperforms RFG when $N=2$.
> - Ablation on strength of reference model is an othogonal contribution, and we've conducted ablation on sensitivity to $w$.
>
> We hope these clarifications address the reviewer’s concerns and welcome further discussion.

---

### Meta-Review · Area_Chair_ii1r · 2026-01-05

**Summary:**

The paper proposes a training-free test-time method for guiding diffusion language model reasoning. The main idea is the guide based on the log-likelihood ratios between the post-trained dLLM and its Base model.  Reviewers agree that the problem of training free test time scaling of diffusion language models is important and relevant, especially given consistent empirical gains across math and coding benchmarks. However, multiple reviewers questioned if the method proposes a fundamental new solution beyond classifier-free guidance, and if test-time scaling is the correct formulation for 2x increase in inference cost. Given the rebuttal discussion results, I recommend the paper for rejection as non of the reviewers had a significantly strong rating.

**Reviewer Concerns:**

The rebuttal addressed concerns about compute-matched baselines, tokenizer constraints, multi-step and multi-token diffusion settings. However, core concerns about limited novelty and practical overhead relative to alternative inference strategies remain outstanding for several reviewers. Unfortunately, only 1 reviewer participated in the rebuttal discussion, and it is not clear if concerns of others are addressed.

**Reviewer Scores:**

Reviewer JGzp would potentially increase the score from 4 -> 6, while others will remain at initial 4.

---

### Decision · Program_Chairs · 2026-01-26

Reject